# Dynamic Stashing Quantization for Efficient Transformer Training

**Guo Yang**
University of Cambridge
Cambridge, UK
gy261@cam.ac.uk

**Robert D. Mullins**
University of Cambridge
Cambridge, UK
robert.mullins@cl.cam.ac.uk

**Daniel Lo**
Microsoft Research
Redmond, Washington, USA
dlo@microsoft.com

**Yiren Zhao**
Imperial College London
London, UK
a.zhao@imperial.ac.uk

## Abstract

Large Language Models (LLMs) have demonstrated impressive performance on a range of Natural Language Processing (NLP) tasks. Unfortunately, the immense amount of computations and memory accesses required for LLM training makes them prohibitively expensive in terms of hardware cost, and thus challenging to deploy in use cases such as on-device learning.

In this paper, motivated by the observation that LLM training is memory-bound, we propose a novel dynamic quantization strategy, termed *Dynamic Stashing Quantization* (DSQ), that puts a special focus on reducing the memory operations, but also enjoys the other benefits of low precision training, such as the reduced arithmetic cost. We conduct a thorough study on two translation tasks (trained-from-scratch) and three classification tasks (fine-tuning). DSQ reduces the amount of arithmetic operations by $20.95\times$ and the number of DRAM operations by $2.55\times$ on IWSLT17 compared to the standard 16-bit fixed-point.

## 1 Introduction

Large Language Models (LLMs) based on the Transformer architectures (Vaswani et al., 2017) are currently seen as the foundation models (Bommasani et al., 2021). The *pre-train and then fine-tune* paradigm has shown promising results for a variety of Natural Language Processing (NLP) tasks (Liu et al., 2019; Raffel et al., 2020; Brown et al., 2020). However, the training of LLM is both computationally and memory intensive, posing a significant challenge for their deployment.

In the hardware world, the *Roofline model* demonstrates that there is an optimal balance of processor and memory performance. The metric used to assess performance is referred to as the *operational intensity*, which is calculated as the ratio of arithmetic intensity to memory bandwidth:

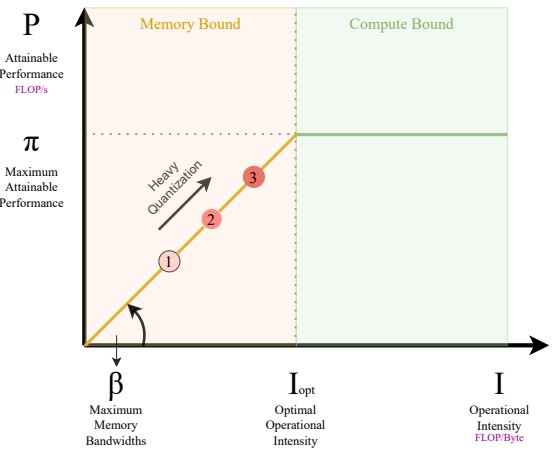

Figure 1: The Roofline model with operational intensity ($I$) and attainable performance ($P$). 1 is non-quantized, 2 is a standard quantization and 3 is DSQ.

$$\text{Operational Intensity} = \frac{\text{Number of Operations}}{\text{DRAM traffic}}$$

The Roofline model has enabled us to identify the sweet spot ($I_{opt}$) for a processor to reach its peak arithmetic performance (Williams et al., 2009; Ding et al., 2022). As illustrated in Figure 1, as operational intensity ($I$) increases, the maximum attainable performance rises at a linear rate initially before reaching a constant value. The region to the left of the turning point is limited by the available memory bandwidth; the region to the right is constrained by the processor's arithmetic computing capability. Training Transformer models, as shown by Ivanov et al. (2021), is memory-bound, which means it sits at the left quadrant in the Roofline model ($I < I_{opt}$). Consequently, the performance of LLM training on modern hardware is significantly hindered by the inadequate bandwidth, as the amount of data movements to and from DRAM is the major performance bottleneck.

For this reason, researchers have sought to accelerate the training process of Transformers through

*quantization.* This approach aims to reduce memory consumption by decreasing the precision of parameters. Prior work has looked into the effect of quantization on Transformer models, a majority of which focus on the forward pass of model inference with fixed weights (Zhang et al., 2020; Bai et al., 2020; Tao et al., 2022). A number of studies have also investigated low-precision training for Transformers (Sun et al., 2019, 2020). Although works have demonstrated the effectiveness of quantization, they typically assume a single precision level, either per neural network layer or per network, which over-simplifies the hardware target. When viewed from a Roofline model perspective, *existing quantization methods attempt to optimize both compute complexity and memory bandwidth requirement, and then fail to recognize that the workload is heavily memory-bound.*

Motivated by this observation, we propose a novel quantization strategy for LLM training named *Dynamic Stashing Quantization* (DSQ). We identify the most memory-intensive part of LLM training – the communication between the forward and backward passes, and define *stashing* as the process of storing intermediate results in a memory buffer (in a normal case, DRAM) for later use. The proposed quantization places an emphasis on this communication, and *dynamically quantize the intermediate results between forward and backward passes* for a significant reduction of the DRAM traffic. As illustrated in Figure 1, this reduction of DRAM bandwidth helps DSQ to move closer to the optimal operational intensity. We have the following contributions:

- We propose *Dynamic Stashing Quantization (DSQ)* for LLM training. DSQ not only quantizes operations for the entire training process, but also employs a *more aggressive quantization* for intermediate results between the forward and backward passes to drastically minimize DRAM traffic.
- DSQ follows a time-adaptive principle for stashing, which involves starting with lower precision at the beginning of the training process and gradually increasing the precision as it progresses. DSQ has been demonstrated to provide a higher performance compared to its fixed-precision counterpart.
- We evaluate the proposed strategy on a variety of tasks and setups, including training from scratch and fine-tuning. DSQ achieves up to a

$2.55\times$ increase in arithmetic performance and a $20.95\times$ reduction in DRAM requirement compared to 16-bit fixed-point training.

## 2   Related Work

Quantization has been studied in detail for inference. These include using uniform (Zafrir et al., 2019; Bhandare et al., 2019) and non-uniform (Sun et al., 2019; Darvish Rouhani et al., 2020) quantization methods. Specifically, uniform quantization methods such as fixed-point (Zafrir et al., 2019; Bhandare et al., 2019; Lin et al., 2020), ternary (Zhang et al., 2020), or even binary (Bai et al., 2020) number formats have been applied to inference of Transformer models. In this work, we focus on quantization for LLM training which introduces new challenges such as the large dynamic range needed during the backward pass for lossless training (Sun et al., 2019) where non-uniform quantization methods have seen more success.

Training LLM models is approximately $3\times$ more expensive than running inference for the same model. Thus, quantizing all operations during training has been an area of active research (Sun et al., 2019, 2020; Yang et al., 2019; Fu et al., 2020; Fox et al., 2020; Kalamkar et al., 2019). Most of these methods use non-uniform quantization to handle larger dynamic range needed for gradient updates (Kalamkar et al., 2019). Floating-point arithmetic and its variants have become a popular method for low-precision training (*e.g.* fewer than 8 bits). Mini-floats with extremely small exponents (*e.g.* 1 bit or 2 bits) have been demonstrated to be effective in small language models, such as LSTMs (Sun et al., 2019, 2020). Block floating-point or block mini-floats, where an exponent is shared between a set of values, has become popular in quantized training (Yang et al., 2019; Drumond et al., 2018; Fox et al., 2020) as it allows for a large dynamic range while approximating the cost of integer formats for multiplication. Specifically, Draumond *et al.* utilized block floating-point with roughly 24 bits to perform lossless training on vision tasks (Drumond et al., 2018). Fox *et al.* demonstrated that 8-bit training is possible with an around $0.5$ BLEU score degradation on machine translation (Fox et al., 2020). Our work extends these formats to Large Language Models, includes quantization of stashed weights, and introduces a dynamic aspect to further reduce the required bit widths. The idea of *stashing* has also been explored before by

Jain et al. (2018), although they only focused on applying lossless encoding methods on single precision numbers (Float16). However, in this paper we show a more aggressive stashing techniques (*e.g.* on average less than 4 bits per number) that is time-adaptive for LLM training. Fractrain (Fu et al., 2020), to our knowledge, is the only work that applied the idea of dynamic quantization on standard training, but was primarily focusing on vision tasks. Our work extends dynamic quantization to encompass stashed values and evaluates these effects on LLMs. Prior research on distributed training has looked at reducing the communication cost (Alistarh et al., 2017; Hönig et al., 2022), where Honig *et al.* also investigate how a time-adaptive quantization would help federated systems to learn. These works focused on device-to-device traffic while our work focuses on reducing DRAM traffic.

## 3 Method

Figure 2 provides a high-level illustration of the DSQ flow. We consider the inputs $x_l$ of a neural network layer with parameters $w_l$, and the output of the layer is $x_{l+1}$. In the backward pass, we consider the partial derivatives $dx_l$ of the input and also the gradient of the weights $dw_l$. Naturally, a single training step requires three GEMMs as illustrated in Figure 2. We illustrate four quantization opportunities in this training step and their effects:

- $q_0$: mainly affects the arithmetic density of forward pass, notice it is possible for $x_l$ and $w_l$ to use different precisions, but this optimization is not the focus of our work.
- $q_1$: affects the DRAM memory bandwidth, one key point in our work is that we show $q_1$ can be different from $q_0$ and in fact can be a very aggressive, dynamic quantization.
- $q_2$: affects mainly the computation complexity of the first GEMM in the backward pass.
- $q_3$: affects the DRAM bandwidth and also the computation complexity of the second GEMM in the backward pass.

In our knowledge, we are the first to systematically illustrate the potential effects, both on compute and off-chip memory bandwidth, of various quantization opportunities within a standard training pass. The two GEMMs in the backward pass can be potentially fused (*e.g.* pipelined), and in that case $dx_l$ does not have to be written to and then read from the DRAM. In our cost model estimation, we use a conservative strategy and assume

this tensor is always flushed to DRAM. In DSQ, we use Block Floating Point as the quantizer for $q_0$, $q_1$, $q_2$ and $q_3$, since this quantizer is shown superior to fixed-point quantization (Darvish Rouhani et al., 2020). We also use a time-adaptive quantization strategy, this means the quantization uses a different quantization level $q_i^t$ for each round $t$ of the training. We design DSQ to monotonically increase $q_i^t$ as a function of $t$ and use the validation loss to inform this increase. This monotonic increase strategy has been proven more effective than other complex scheduling methods in Hönig et al. (2022). Through extensive tuning and experimentation, we also notice that it is important to keep $q_3 \geq 16$ through the entire training process, and Appendix C studies the effect of different quantization levels for $q_3$.

## 4 Evaluation

We evaluate the effectiveness of DSQ on two different translation tasks, WMT14 EN-DE (Bojar et al., 2014) (in Appendix D) and IWSLT17 EN-DE (Cettolo et al., 2017), and two tasks from the GLUE benchmark (Wang et al., 2018), the details of these datasets are in Appendix A. We used the Adam optimizer and the details for all the learning rate and batch size selections are in Appendix B. For the translation tasks, we use a classic 6-layer transformer model (Vaswani et al., 2017) and the RoBERTa-base model (Liu et al., 2019) for the GLUE tasks. All tasks are executed on systems that have 2 AMD EPYC 7763 64-Core Processors 1.8GHz (128 cores in total), and 4 NVIDIA A100-SXM-80GB GPUs, with 1000 GiB RAM. We are interested in understanding the costs of arithmetic operations, as well as the number of memory reads and writes. To this end, we have built a hardware performance modeling framework to estimate the training cost. Our cost model is similar to Sun et al. (2020) and Samajdar et al. (2018), but our numbers are derived from a production hardware system, taking the numbers reported in Darvish Rouhani et al. (2020), to provide a higher-fidelity estimation.

Table 1 presents the results of our study comparing different quantization strategies. We compare popular low-latency training baselines and Block floating-point (BFP) (Darvish Rouhani et al., 2020; Fox et al., 2020) with different precisions. For all BFP implementations considered in this paper, we keep the exponent bitwidth to be 8 and the bounding-box size to be 16 following

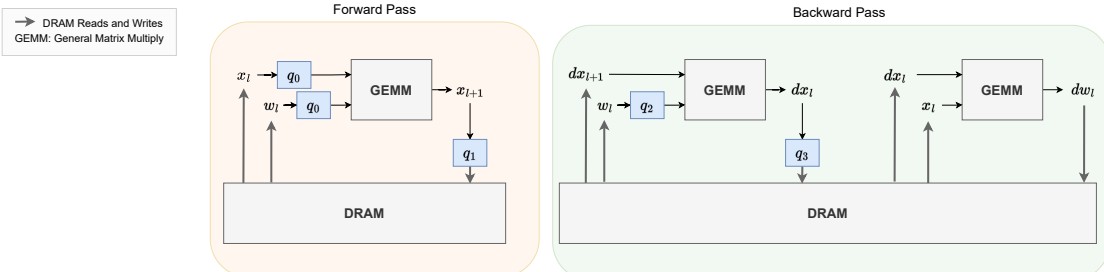

Figure 2: An illustration of the DSQ flow for a single linear layer. The training is viewed as a combination of a forward pass and a backward pass. $q_0, q_1, q_2$ and $q_3$ define where the tensors are quantized, we use $[q_0, q_1, q_2, q_3]$ to describe the DSQ configuration. DSQ ensures all GEMM inputs are quantized. Notice for the second and third GEMMs, $dx_{l+1}$, $x_l$ and $dx_l$ are the quantized version fetched from the DRAM, the fact that these values are heavily quantized helps us to save DRAM bandwidth.

Table 1: The performance of Machine Translation trained with a 6-layer Transformer architecture, the model is assessed using numbers reported as percentages. $\Delta$ shows the performance difference compared to the floating-point 32-bit baseline.

| Dataset and Model | Method | Precision Setup | Acc / BLEU ($\Delta$) | Arith Ops ($\downarrow$) | DRAM R/W ($\downarrow$) |
|---|---|---|---|---|---|
| IWSLT2017 DE-EN Transformer (6-layer) | Floating-point | [32, 32, 32, 32] | 35.22 | - | - |
| | Fixed-point | [32, 32, 32, 32] | 34.47 ($-0.75$) | $1.00\times$ | $1.00\times$ |
| | Fixed-point | [16, 16, 16, 16] | 32.59 ($-2.63$) | $0.25\times$ | $0.50\times$ |
| | Block FP | [32, 32, 32, 32] | 34.56 ($-0.66$) | $0.56\times$ | $1.13\times$ |
| | Block FP | [16, 16, 16, 16] | 34.30 ($-0.92$) | $0.18\times$ | $0.63\times$ |
| | Stashing (Fixed) | [16, 4, 4, 16] | 25.50 ($-9.72$) | $0.13\times$ | $0.31\times$ |
| | Stashing (BFP) | [16, 4, 4, 16] | 34.78 ($-0.44$) | $0.10\times$ | $0.45\times$ |
| | DSQ (BFP) | $-$ | 34.81 ($-0.41$) | $0.012\times$ | $0.20\times$ |
| GLUE MNLI RoBERTa-base | Floating-point | [32, 32, 32, 32] | 87.6 | - | - |
| | Fixed-point | [32, 32, 32, 32] | 87.9 ($+0.3$) | $1.00\times$ | $1.00\times$ |
| | Fixed-point | [16, 16, 16, 16] | 87.9 ($+0.3$) | $0.25\times$ | $0.50\times$ |
| | Block FP | [32, 32, 32, 32] | 87.8 ($+0.2$) | $0.56\times$ | $1.13\times$ |
| | Block FP | [16, 16, 16, 16] | 87.8 ($+0.2$) | $0.18\times$ | $0.63\times$ |
| | Stashing (Fixed) | [16, 4, 4, 16] | 82.8 ($-4.8$) | $0.13\times$ | $0.32\times$ |
| | Stashing (BFP) | [16, 4, 4, 16] | 87.8 ($+0.2$) | $0.10\times$ | $0.45\times$ |
| | DSQ (BFP) | $-$ | 87.8 ($+0.2$) | $0.043\times$ | $0.26\times$ |
| GLUE QNLI RoBERTa-base | Floating-point | [32, 32, 32, 32] | 92.8 | - | - |
| | Fixed-point | [32, 32, 32, 32] | 92.6 ($-0.2$) | $1.00\times$ | $1.00\times$ |
| | Fixed-point | [16, 16, 16, 16] | 92.6 ($-0.2$) | $0.25\times$ | $0.50\times$ |
| | Block FP | [32, 32, 32, 32] | 92.7 ($-0.1$) | $0.56\times$ | $1.13\times$ |
| | Block FP | [16, 16, 16, 16] | 92.5 ($-0.3$) | $0.18\times$ | $0.63\times$ |
| | Stashing (Fixed) | [16, 4, 4, 16] | 89.5 ($-3.3$) | $0.13\times$ | $0.32\times$ |
| | Stashing (BFP) | [16, 4, 4, 16] | 92.6 ($-0.2$) | $0.10\times$ | $0.45\times$ |
| | DSQ (BFP) | $-$ | 92.7 ($-0.1$) | $0.043\times$ | $0.26\times$ |

Darvish Rouhani et al. (2020). In addition, we compare static stashing strategies that are based on either fixed-point (Fixed) or BFP. In Table 1, we use the hardware cost of fixed-point 32-bit computation as $1\times$ since this is a stronger baseline. The results demonstrate that DSQ has a comparable accuracy and BLEU score compared to 16-bit fixed-point for training while having a $20.95\times$ reduction in arithmetic complexity and a $2.55\times$ decrease in DRAM R/W. DSQ also shows very competitive accuracy on fine-tuning RoBERTa on GLUE while having a much smaller hardware utilization.

## 5 Conclusion

In this paper, we propose Dynamic Stashing Quantization (DSQ) for LLM training. This new quantization strategy applies a more aggressive quantization for intermediate results between the forward and backward passes generated during training, thereby reducing DRAM traffic. Specifically, our approach uses a low precision at the beginning of training, and then gradually increases the precision level, to reduce the effect of round-off errors introduced by quantization. We demonstrate the effectiveness of DSQ by showing how it can reduce both the computation cost and DRAM bandwidth requirement on machine translation and LLM fine-tuning tasks.

# 6 Limitation

- DSQ precision configurations are decided through experimentation on the IWSLT dataset. The precision for different stages is scheduled based on the validation loss value, the precision would increase if the validation loss becomes 'flat' (non-decreasing). In our particular method, if our validation loss has been non-decreasing for a fixed number of $N$ epochs, we then move to the next quantization level, following a setup similar to that proposed by Hönig et al. (2022). We find that setting $N = 5$ is sufficient for all test scenarios. The choice of this free parameter is a limitation in our paper and will be investigated in further research. The same precision configuration setup is used for all other datasets.

- Language models that are larger than classic Transformer and RoBERTa have been developed in recent works. Due to limited resouces we have, we choose to work on smaller models as an exemplar. While we have tested on larger LMs like OPT-1.3B (in Appendix E) to show large LLMs today are also memory-bound, additional experimentation is expected to be conducted to enhance the robustness and precision of our findings.

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

## A  Datasets

Four datasets are used: translation WMT14 EN-DE and IWSLT2017 EN-DE for machine translation tasks, QNLI and MNLI for textual entailment tasks. Table 2 presents details for the datasets.

## B  Hyperparameters

The training hyperparamters, such as learning rates, are picked following standard benchmarks and open implementaitons (Liu et al., 2019; Vaswani et al., 2017). We summarize them in Table 3 for repeatability. We use the Adam optimizer with $\beta_1 = 0.9$, $\beta_2 = 0.98$ for both training and finetuning models. The learning rate schedule is Inverse Square Root for training models, and Polynomial Decay for finetuning models. Dropout with rates of $P_{IWSLT} = 0.3$ and $P_{WMT} = 0.2$, label smoothing with value $\epsilon = 0.1$ are applied to train models.

DSQ precision configurations are decided through experimentation on the IWSLT dataset as discussed in section 6 Limitations. Table 4 shows a collection of tuning runs we had, we found that heavily quantized models still work at the start of training stage, and [16, 4, 4, 16] quantized BFP model works as well as less aggressive ones. This indicates that DSQ should start with heavily aggressive precision setup (we pick [2, 2, 2, 16] for IWSLT14 DE-EN), and jump to [16, 4, 4, 16] when needed during training process.

Table 2: Details for each dataset, including the number of classes, a description and the source.

| Name | # Class | Description |
|---|---|---|
| WMT14 EN-DE | - | A text translation task on English-German sentence pairs from The The Stanford Natural Language Processing Group. |
| IWSLT2017 DE-EN | - | A text translation task on German-English sentence pairs from The International Conference on Spoken Language Translation. |
| QNLI | 2 | A binary textual entailment task on question-answer pairs from the Stanford Question Answering database. The objective is to determine whether a pair is an entailment or not. |
| MNLI | 3 | A multi-class (i.e., entailment, neutral, contradiction) textual entailment task on premise-hypothesis pairs from the Multi-genre Natural Language Inference corpus. Matched version only preserves pairs within the same genre (e.g., government report, science fiction, speech). |

Table 3: Details of the optimal hyper-parameters including batch size, learning rate and weight decay values for each set of experiments with the same dataset and prompting model.

| Dataset | Batch size | Max tokens | Learning rate | Weight decay |
|---|---|---|---|---|
| WMT14 EN-DE | - | 4096 | 5e-4 | 0.0 |
| IWSLT2017 DE-EN | - | 4096 | 5e-4 | 1e-4 |
| QNLI | 32 | 4400 | 1e-5 | 0.1 |
| MNLI | 32 | 4400 | 1e-5 | 0.1 |

Table 4: Tests on stashing precision setup. The models are trained on IWSLT14 DE-EN. $\Delta$ shows the performance difference compared to the floating-point 32-bit baseline.

| Dataset and Model | Method | Precision Setup | Acc / BLEU ($\Delta$) |
|---|---|---|---|
| IWSLT14 DE-EN Transformer (6-layer) | Stashing (BFP) | [2, 2, 2, 16] | 17.45 ($-17.77$) |
| | Stashing (BFP) | [4, 2, 2, 16] | 33.51 ($-1.71$) |
| | Stashing (BFP) | [4, 4, 4, 16] | 34.47 ($-0.75$) |
| | Stashing (BFP) | [8, 4, 4, 16] | 34.47 ($-0.75$) |
| | Stashing (BFP) | [8, 8, 8, 16] | 34.65 ($-0.57$) |
| | Stashing (BFP) | [16, 4, 4, 16] | 34.78 ($-0.44$) |
| | Stashing (BFP) | [16, 8, 8, 16] | 34.47 ($-0.75$) |

## C  The effect of $q_3$

The gradient output ($dx_l$) plays an important role in the performance of fixed-point quantization. Notice in table Table 5, gradient output quantized to 8 bits leads to training failure for fixed-point quantization. In order to focus on the idea of stashing, we apply 16 bits quantization of gradient output for all our stashing precision setups.

## D  Additional results on WMT14

We also train the model on WMT14 EN-DE dataset, the BLEU scores we gain are relatively low compared to the 27.3 BLEU score achieved by Vaswani et al. (2017) because we only trained the models for 15 epochs. Table 6 presents the results.

## E  Memory-bound verification on OPT-1.3B

we have also tested on larger LMs like OPT-1.3B to show large LLMs today are also memory-bound so that dynamic stashing will serve as a useful technique for reducing the DRAM traffic and compute. We run a similar analysis to Ivanov et al. (2021). Table 7 presents the results. The analysis agrees with the conclusion of Ivanov et al. (2021). This may be of interest to the community in understanding the memory-bound nature of current LLMs.

Table 5: Tests on gradient output precision setup. The models are trained on IWSLT14 DE-EN.

| Dataset and Model | Method | Precision Setup | Acc / BLEU ($\Delta$) |
|---|---|---|---|
| IWSLT14 DE-EN
Transformer (6-layer) | Stashing (Fixed) | [8, 8, 8, 32] | 34.08 |
| | Stashing (Fixed) | [8, 8, 8, 16] | 31.94 |
| | Stashing (Fixed) | [8, 8, 8, 8] | Failed |

Table 6: The performance of Machine Translation trained on WMT14 EN-DE with a 6-layer Transformer architecture (Vaswani et al., 2017), the model is assessed using numbers reported as percentages. $\Delta$ shows the performance difference compared to the floating-point 32-bit baseline.

| Dataset and Model | Method | Precision Setup | Acc / BLEU ($\Delta$) | Arith Ops | DRAM R/W |
|---|---|---|---|---|---|
| WMT14 EN-DE
Transformer (6-layer) | Floating-point | [32, 32, 32, 32] | 25.79 | - | - |
| | Fixed-point | [32, 32, 32, 32] | 25.41 ($-0.38$) | 1.00× | 1.00× |
| | Fixed-point | [16, 16, 16, 16] | 23.40 ($-2.39$) | 0.25× | 0.50× |
| | Block FP | [32, 32, 32, 32] | 25.76 ($-0.03$) | 0.56× | 1.13× |
| | Block FP | [16, 16, 16, 16] | 25.61 ($-0.18$) | 0.18× | 0.63× |
| | Stashing (Fixed) | [16, 4, 4, 16] | 21.86 ($-3.93$) | 0.13× | 0.31× |
| | Stashing (BFP) | [16, 4, 4, 16] | 25.24 ($-0.55$) | 0.10× | 0.20× |

Table 7: Bound type analysis of OPT-1.3B based on computational complexity and latency fraction

| Model | | Attention Layers | FC Layers | Others |
|---|---|---|---|---|
| OPT-1.3B | Latency Fraction | 47.93% | 32.20% | 19.87% |
| | Bound Type | Memory-bound | Compute-bound | Memory-bound |