# OpenReview forum: "Dynamic Stashing Quantization for Efficient Transformer Training"
_EMNLP/2023/Conference — EMNLP 2023 Findings_

### Official Review · Reviewer_Qa9X · 2023-08-03

**Soundness:** 4

**Excitement:**

4: Strong: This paper deepens the understanding of some phenomenon or lowers the barriers to an existing research direction.

**Paper Topic And Main Contributions:**

This paper proposes a novel dynamic quantization strategy, termed Dynamic Stashing Quantization (DSQ) to reduce the amount of computations and memory accesses required for LLM training.

**Reasons To Accept:**

1) The motivation is great. Reducing memory consumption and FLOPs is very important for LLM training efficiency.
2) The results are very impressive. It is good to write such a concise idea in a short paper.

**Reasons To Reject:**

1) The model is too small. I understand the resource is a problem. Authors do not need to fix this during rebuttal. But I have to say this as a limitation.
2) Are there any training stability issue? Since the model used is small, the training stability is not a big issue. But I believe it would be beneficial to try a larger model with higher learning rate to check how stable the approach is.

**Reproducibility:**

3: Could reproduce the results with some difficulty. The settings of parameters are underspecified or subjectively determined; the training/evaluation data are not widely available.

**Reviewer Confidence:**

3: Pretty sure, but there's a chance I missed something. Although I have a good feel for this area in general, I did not carefully check the paper's details, e.g., the math, experimental design, or novelty.

---

> ### Author Rebuttal · Authors · 2023-08-28
>
> We thank the reviewer for the comments.
>
> The compute resource is a high-performance computing facility that is shared among a number of researchers. Due to the limited compute hours available to us, we have chosen to work on smaller models as an exemplar. Nevertheless, in Appendix E, We run a similar analysis to Ivanov et al. on larger LMs like OPT-1.3B to show LLMs today are also memory-bound, so that DSQ will serve as a useful technique for reducing the DRAM traffic and compute.
>
> We thank the reviewer for the comments regarding training stability. Empirically, we observe that our training log figures show stable gradients and loss. We will include the figures in our final draft.

---

### Official Review · Reviewer_3Ppe · 2023-08-05

**Soundness:** 3

**Excitement:**

4: Strong: This paper deepens the understanding of some phenomenon or lowers the barriers to an existing research direction.

**Paper Topic And Main Contributions:**

This paper addresses the challenges associated with training Large Language Models (LLMs), particularly focusing on the high computational and memory demands that make such models expensive and difficult to deploy in situations like on-device learning. The authors propose Dynamic Stashing Quantization (DSQ), a novel dynamic quantization strategy designed to significantly reduce memory operations while also benefiting from the reduced arithmetic cost associated with low precision training. The method's efficacy is demonstrated through a comprehensive study on two translation tasks (trained-from-scratch) and three classification tasks (fine-tuning).

**Questions For The Authors:**

How does DSQ compare to other existing quantization strategies in terms of efficiency and performance?

**Reasons To Accept:**

1. The paper tackles an important problem in the realm of LLM training, namely the high computational and memory requirements.
2. The idea of time-adaptive compression, which involves starting with lower precision at the beginning of the training process and gradually increasing the precision as it progresses, is interesting and novel.
3. The authors provide a sufficient evaluation of DSQ, including tests on both translation and classification tasks. The evaluation adds to the paper's credibility and provides a robust demonstration of DSQ's effectiveness.

**Reasons To Reject:**

1. The paper primarily focuses on smaller-scale models, leaving some uncertainty about the applicability and effectiveness of the proposed DSQ method on more complex, large-scale models. It would significantly enhance the paper's value if the authors could extend their experiments to include fine-tuning of larger models, such as OPT 1.3B and LLaMA, and provide a comparative analysis of the perplexity outcomes.
2. The paper provides a comparison of arithmetic complexity and DRAM Read and Writes, but it would be great to include an end-to-end speedup analysis, which would offer a more holistic understanding of DSQ's performance in real-world scenarios.

**Reproducibility:**

4: Could mostly reproduce the results, but there may be some variation because of sample variance or minor variations in their interpretation of the protocol or method.

**Reviewer Confidence:**

3: Pretty sure, but there's a chance I missed something. Although I have a good feel for this area in general, I did not carefully check the paper's details, e.g., the math, experimental design, or novelty.

---

> ### Author Rebuttal · Authors · 2023-08-28
>
> We thank the reviewer for the comments.
>
> The compute resource is a high-performance computing facility that is shared among a number of researchers. Due to the limited compute hours available to us, we have chosen to work on smaller models as an exemplar. Nevertheless, in Appendix E, We run a similar analysis to Ivanov et al. on larger LMs like OPT-1.3B to show LLMs today are also memory-bound, so that DSQ will serve as a useful technique for reducing the DRAM traffic and compute.
>
> It is worth noting that there is a growing trend towards utilizing domain-specific hardware for accelerating ML workloads, exemplified by Google’s TPU for AI acceleration as compared to NVIDIA’s GPGPU. In this regard, we hold the belief that ASICs optimized for efficiency and cost will be the future generation hardware for language model training.
>
> NVIDIA’s GPU is the most common choice for training and testing neural networks in the deep learning community because deep learning frameworks like PyTorch have high-performance tensor computation libraries built on CUDA e.g., Aten for PyTorch. However, one limitation of GPUs is the lack of native support for custom quantization arithmetics like the BFP mentioned in this paper. For example, NVIDIA RTX 4090 only supports FP64, FP32, Bfloat16, FP16, and Int8, and PyTorch supports training & testing with FP64, FP32, Bfloat16, and FP16. Though sometimes it is possible to support custom quantization arithmetics via manually-designed CUDA kernel, the efficiency is still much lower than ASICs.
>
> In conclusion, we are exploring next-generation hardware for training language models, which requires the implementation of DSQ on specialised hardware in order to achieve a speed-up. We will revise our draft to include this to our Appendix.
>
> Our answer to the reviewer's question:
>
> We noticed that DRAM bandwidth is a more critical limiting factor for Transformer training than for training vision networks. Most existing quantization methods treat all components in the network equally and utilise mainly fixed-point quantization. To combat this, we implemented both fixed-point quantization and BFP quantization, and found out the BFP performed better in Transformer setup. We then proposed a dynamic (time-adaptive) stashing technique with BFP (DSQ) to reduce DRAM traffic. The precision of DSQ can go up to 32 bits, providing us with a nice fall-back scenario where the accuracy can match that of large bitwidth quantization methods.

---

### Official Review · Reviewer_vpzW · 2023-08-10

**Typos Grammar Style And Presentation Improvements:** 1. A nit, talking about LLMs in Intro…
**Soundness:** 3

**Excitement:**

3: Ambivalent: It has merits (e.g., it reports state-of-the-art results, the idea is nice), but there are key weaknesses (e.g., it describes incremental work), and it can significantly benefit from another round of revision. However, I won't object to accepting it if my co-reviewers champion it.

**Missing References:**

1. There has been a lot of work in the domain of quantization of gradients used for synchronization across DP workers. The work in this domain should go in related work section

**Paper Topic And Main Contributions:**

This paper focuses on a dynamic quantization algorithm thats majorly focussed on reducing the DRAM traffic during training and hence improve the operational intensity of the problem. They improve upon the usual quantization flow by identifying that the quantization of weights and activations during the forward and backward can be different. Especially, the activations being read for gradient calculations can be significantly quantized. The algorithm focusses on 4 different quantization operations - activations and weights during forward calculation, output activation after forward call being written to DRAM for gradient calculation, activations being read for gradient calculations and weight quantization during gradient calculation. The quantization level changes as training progresses.

**Questions For The Authors:**

1. What is the time to train for the baseline models evaluated in this paper?
2. How does this technique interact with gradient checkpointing, replays and gradient accumulation? Standard techniques used from LLM training.

**Reasons To Accept:**

1. Paper is easy to ready and understand
2. The focus on quantization of gradient communication between forward and backward is interesting.

**Reasons To Reject:**

1. The method while interesting needs a lot more benchmarking on more complex tasks to ensure that it works generally. Pushing for a larger model like BERT will also help. For finetuning tasks like SQuAD will also add credence to the work.

2. Without any runtime numbers, its hard to understand the wall clock impact of these techniques. Additionally, arithmetic Ops reduction will not necessary translate to faster runtime for arbitrary quantization bit-widths. As a result, the impact of this work may be limited.

**Reproducibility:**

3: Could reproduce the results with some difficulty. The settings of parameters are underspecified or subjectively determined; the training/evaluation data are not widely available.

**Reviewer Confidence:**

4: Quite sure. I tried to check the important points carefully. It's unlikely, though conceivable, that I missed something that should affect my ratings.

---

> ### Author Rebuttal · Authors · 2023-08-28
>
> We thank the reviewer for the comments.
>
> The compute resource is a high-performance computing facility that is shared among a number of researchers. Due to the limited compute hours available to us, we have chosen to work on smaller models as an exemplar. Nevertheless, in Appendix E, We run a similar analysis to Ivanov et al. on larger LMs like OPT-1.3B to show LLMs today are also memory-bound, so that DSQ will serve as a useful technique for reducing the DRAM traffic and compute.
>
> Our answers to the reviewer's questions:
>
> Question 1:
>
> It is worth noting that there is a growing trend towards utilizing domain-specific hardware for accelerating ML workloads, exemplified by Google’s TPU for AI acceleration as compared to NVIDIA’s GPGPU. In this regard, we hold the belief that ASICs optimized for efficiency and cost will be the future generation hardware for language model training.
>
> NVIDIA’s GPU is the most common choice for training and testing neural networks in the deep learning community because deep learning frameworks like PyTorch have high-performance tensor computation libraries built on CUDA e.g., Aten for PyTorch. However, one limitation of GPUs is the lack of native support for custom quantization arithmetics like the BFP mentioned in this paper. For example, NVIDIA RTX 4090 only supports FP64, FP32, Bfloat16, FP16, and Int8, and PyTorch supports training & testing with FP64, FP32, Bfloat16, and FP16. Though sometimes it is possible to support custom quantization arithmetics via manually-designed CUDA kernel, the efficiency is still much lower than ASICs.
>
> In conclusion, we are exploring next-generation hardware for training language models, which requires the implementation of DSQ on specialised hardware in order to achieve a speed-up. We will revise our draft to include this to our Appendix.
>
> Question 2:
>
> DSQ operates as a quantization methodology and does not inherently change the way of using gradient checkpointing, replays and gradient accumulation. In other words, these techniques can be applied in conjunction to DSQ. However, it is worth noting that the precision of DSQ changes during training, and makes less utilisation of the memory. In this case, if applying gradient checkpointing/replay, we should in theory see less layers being recomputed in the backward pass. Moreover, this method operates in a separate optimization space when it comes to gradient accumulation.

---

### Official Review · Reviewer_jF2i · 2023-08-11

**Soundness:** 3

**Excitement:**

4: Strong: This paper deepens the understanding of some phenomenon or lowers the barriers to an existing research direction.

**Missing References:**

None

**Paper Topic And Main Contributions:**

Dynamic Stashing Quantization (DSQ) is introduced as a novel approach for Language Model (LLM) training. It involves quantizing operations throughout training and emphasizes aggressive quantization for intermediate results between forward and backward passes to minimize DRAM traffic. DSQ employs a time-adaptive stashing principle, starting with lower precision and gradually increasing it during training. This technique outperforms fixed-precision methods, resulting in significant improvements in performance. Evaluation across various tasks and setups, including training from scratch and fine-tuning, demonstrates DSQ's effectiveness. It achieves up to a 2.55× boost in arithmetic performance and reduces DRAM requirements by 20.95× compared to 16-bit fixed-point training.

**Reasons To Accept:**

The concept of Dynamic Stashing Quantization (DSQ) is innovative and its emphasis on aggressive quantization for intermediate results is well-motivated. The time-adaptive stashing principle introduces a dynamic element to the quantization process, which is a notable contribution.

The experimental evaluation is thorough, encompassing various tasks and scenarios, and showcasing DSQ's superiority over fixed-precision methods. The achieved performance improvements are impressive, with up to a 2.55× boost in arithmetic performance and a substantial 20.95× reduction in DRAM requirements compared to 16-bit fixed-point training.





**Reasons To Reject:**

While the paper's content is solid, I believe some sections could benefit from further elaboration. Specifically, more details on the practical implementation of DSQ, as well as potential challenges and limitations, would enhance the paper's completeness.

**Reproducibility:**

3: Could reproduce the results with some difficulty. The settings of parameters are underspecified or subjectively determined; the training/evaluation data are not widely available.

**Reviewer Confidence:**

2: Willing to defend my evaluation, but it is fairly likely that I missed some details, didn't understand some central points, or can't be sure about the novelty of the work.

---

> ### Author Rebuttal · Authors · 2023-08-28
>
> We thank the reviewer for the comments.
>
> It is worth noting that there is a growing trend towards utilizing domain-specific hardware for accelerating ML workloads, exemplified by Google’s TPU, AMD’s AI Engine and Cerebras WSE for AI acceleration as compared to NVIDIA’s GPGPU. In this regard, we hold the belief that ASICs optimized for efficiency and cost will be the future generation hardware for language model training.
>
> NVIDIA’s GPU is the most common choice for training and testing neural networks in the deep learning community because deep learning frameworks like PyTorch have high-performance tensor computation libraries built on CUDA e.g., Aten for PyTorch. However, one limitation of GPUs is the lack of native support for custom quantization arithmetics like the BFP mentioned in this paper. For example, NVIDIA RTX 4090 only supports FP64, FP32, Bfloat16, FP16, and Int8, and PyTorch supports training & testing with FP64, FP32, Bfloat16, and FP16. Though sometimes it is possible to support custom quantization arithmetics via manually-designed CUDA kernel, the efficiency is still much lower than ASICs.
>
> In conclusion, we are exploring next-generation hardware for training language models, and the implementation of DSQ requires specialised hardware. Potential challenges may include hardware support and existing algorithm adaptation. We will add this explanation to our revised version of the paper.

---

### Meta-Review · Area_Chair_MyXF · 2023-09-08

**Recommendation:** 5

**Metareview:**

When I look at this paper, I see some flaws and statements that will not translate to real hardware and personally, I find it a bit misleading. However, the overall response from the reviewers is strongly in factor of acceptance both in soundness and excitement, and I will not overrule their decision. As such, I recommend acceptance to the main conference.

---

### Decision · Program_Chairs · 2023-10-07

**Decision:**

Accept-Findings

**Comment:**

When I look at this paper, I see some flaws and statements that will not translate to real hardware and personally, I find it a bit misleading. However, the overall response from the reviewers is strongly in factor of acceptance both in soundness and excitement, and I will not overrule their decision. As such, I recommend acceptance to the main conference.